# Ligand-induced activation and G protein coupling of prostaglandin F$_{2\alpha}$ receptor

Canrong Wu [1,7] ✉, Youwei Xu [1,7], Qian He[1], Dianrong Li[2], Jia Duan[1], Changyao Li[3,4], Chongzhao You [1], Han Chen [5], Weiliang Fan[2], Yi Jiang[3,4] & H. Eric Xu [1,6] ✉

Prostaglandin F$_{2\alpha}$ (PGF$_{2\alpha}$), an endogenous arachidonic acid metabolite, regulates diverse physiological functions in many tissues and cell types through binding and activation of a G-protein-coupled receptor (GPCR), the PGF$_{2\alpha}$ receptor (FP), which also is the primary therapeutic target for glaucoma and several other diseases. Here, we report cryo-electron microscopy (cryo-EM) structures of the human FP bound to endogenous ligand PGF$_{2\alpha}$ and anti-glaucoma drugs LTPA and TFPA at global resolutions of 2.67 Å, 2.78 Å, and 3.14 Å. These structures reveal distinct features of FP within the lipid receptor family in terms of ligand binding selectivity, its receptor activation, and G protein coupling mechanisms, including activation in the absence of canonical PIF and ERY motifs and G$_q$ coupling through direct interactions with receptor transmembrane helix 1 and intracellular loop 1. Together with mutagenesis and functional studies, our structures reveal mechanisms of ligand recognition, receptor activation, and G protein coupling by FP, which could facilitate rational design of FP-targeting drugs.

Prostanoids are a class of oxygenated arachidonic acid metabolites that include prostaglandin F$_{2\alpha}$ (PGF$_{2\alpha}$), prostaglandin D$_2$ (PGD$_2$), prostaglandin E$_2$ (PGE$_2$), thromboxane A$_2$ (TXA$_2$), and prostacyclin (PGI$_2$). They provoke diverse biological actions in many tissues and cell types through direct interactions with nine lipid G-protein-coupled receptors (GPCRs), prostaglandin F$_{2\alpha}$ receptor (FP), prostaglandin D$_2$ receptors (DP1-2), prostaglandin E$_2$ receptors (EP1-4), thromboxane receptor (TP) and prostacyclin receptor (IP1)[1], which comprise a subfamily of class A GPCRs. FP is encoded in humans by the PTGFR gene[2]. Stimulated by PGF$_{2\alpha}$, FP plays a pivotal role in regulating inflammation, allergic responses, intraocular pressure, and blood pressure, making it a valuable target for therapeutic discovery and development[1,3–5]. FP is highly expressed in uterine myometrium, eye, smooth muscle, skin, and ovarian[6,7]. Upon agonist stimulation, FP is predominantly coupled to the G$_q$ subtype of G proteins, which activation leads to subsequent PKC

activation and a transient calcium signaling in response to the formation of inositol triphosphate[8,9]. In addition to G$_q$, FP activation also induces activation of small G protein Rho via G$_{12}$/G$_{13}$[10] and activation of Raf/MEK/MAP kinase pathway through Gi[11].

Given the diverse functions of FP, it has been targeted for therapeutic development. PGF$_{2\alpha}$, the endogenous FP activator, entered a clinical trial for glaucoma treatment[12]. However, its clinical application was limited by intolerable side effects, possibly caused by its low selectivity for FP receptor[12]. Since then, selective FP agonists have attracted extensive attention and have been developed for the treatment of glaucoma[5], scalp alopecia[13], and vitiligo[14]. From 1996 to 2012, several FP-selective prostaglandin analogs (PGAs) were approved by the United States Food and Drug Administration (FDA) for glaucoma treatment. As a prodrug of a selective FP receptor agonist, latanoprost (LTP) was approved to treat glaucoma for the first time in 1996. It has

[1]State Key Laboratory of Drug Research, Shanghai Institute of Materia Medica, Chinese Academy of Sciences, Shanghai 201203, China. [2]Sironax (Beijing) Co., Ltd., Beijing 102206, China. [3]Lingang Laboratory, Shanghai 200031, China. [4]School of Life Science and Technology, ShanghaiTech University, 201210 Shanghai, China. [5]Department of Biochemistry and Molecular Biology, School of Basic Medical Sciences, Fujian Medical University, Fuzhou, Fujian 350108, China. [6]University of Chinese Academy of Sciences, Beijing 100049, China. [7]These authors contributed equally: Canrong Wu, Youwei Xu. ✉e-mail: wucanrong@simm.ac.cn; eric.xu@simm.ac.cn

also been used to treat scalp alopecia[15] and vitiligo[14] in recent years. Latanoprost acid (LTPA), 17-phenyl-13,14-dihydro PGF$_{2\alpha}$, is an active metabolic form of LTP. Another fluorinated PGA prodrug, tafluprost (TFP), was first approved for the treatment of glaucoma in 2012[16]. Tafluprost acid (TFPA), 15-deoxy-15,15-difluoro-16-phenoxy PGF$_{2\alpha}$, is the active metabolic form of TFP. Due to their high efficacy, these PGAs have been the first-line drug in clinics for the treatment of glaucoma. However, through post-marketing surveillance, 5–20% of patients suffered side effects such as conjunctival congestion and headache, including intolerance responses toward these PGAs[17–19]. Improving the selectivity of these drugs to the FP receptor and elucidating the molecular mechanisms underlying the functional selectivity of individual prostanoid receptor family members are highly important and clinically relevant.

Extensive efforts have been made to clarify how the binding of endogenous and synthetic ligands with various pharmacological profiles regulate FP's downstream signaling[8–11]. However, the molecular details defining the binding modes of ligands remain largely unknown, which is partly attributed to the scarcity of the structural information on ligands bound FP complex. Understanding the mechanism of prostaglandin-FP signaling and identifying differences in the ligand selectivity of prostaglandin receptors may assist in the development of selective drugs with improved safety.

Here we present three cryo-EM structures of G protein-coupled FP in complex with its endogenous ligand PGF$_{2\alpha}$ and with two synthetic agonists, LTPA and TFPA, at global resolutions of 2.67, 2.78, and 3.14 Å. Combined with functional characterizations of mutated receptors, these structures reveal conserved and divergent mechanisms of ligand binding, receptor activation, and G protein coupling by FP.

## Results

### Cryo-EM analysis and overall structure

To facilitate the expression of FP complexes, we introduced a BRIL tag to the N-terminus of the wild-type (WT) receptor[20,21]. A G$\alpha_q$ chimera was engineered based on the mini-G$\alpha_s$ scaffold with an N-terminal replacement of corresponding sequences of G$\alpha_{i1}$ to facilitate the binding of scFv16[22–24]. Hereinafter, G$\alpha_q$ reference to G$\alpha_q$ chimera. The FP-G$_q$ complex was further stabilized by the NanoBiT strategy[25]. Incubation of PGF$_{2\alpha}$/LTPA/TFPA with membranes from cells co-expressing receptors and heterotrimer G$_q$ proteins in the presence of scFv16 and Nb35 enables efficient assembly of the PGF$_{2\alpha}$/LTPA/TFPA -FP-G$_q$ complexes, which produces highly homogenous complex samples for structural studies[26] (Supplementary Figs. 1–3, Table 1). The structures of the FP-G$_q$-scFv16-Nb35 complexes with PGF$_{2\alpha}$, LTPA, and TFPA were determined by cryo-EM to the resolutions of 2.67, 2.78, and 3.14 Å (Fig. 1a, Supplementary Figs. 1–4). The high-quality density map allowed unambiguous model building for the receptor structure containing residues 29–323, except for two invisible residues in the intracellular loop 3 (ICL3) (residues 238 and 239). The density maps are also clear for three agonists, most residues of the G$_q$ heterotrimer, scFv16, and Nb35 (Fig. 1b, c Supplementary Fig. 4).

The overall structure of the active FP receptor is highly similar to those of active EP2 (PDB code: 7CX2) and EP4 (PDB code: 7D7M), with root mean square deviation (RMSD) values of 1.27 and 1.23 Å, respectively. FP folds into a canonical seven-transmembrane helical domain (TMD). All three extracellular loops (ECLs 1–3) were well defined, where ECL2 forms a β-hairpin loop, which is stabilized by the highly conserved disulfide bond between C186$^{ECL2}$ and C108$^{3.25}$. The β-hairpin ECL2 of FP resembles those of EP2 and EP4, and tightly caps the extracellular region (Fig. 1d). We superpose the TMD structures of PGF$_{2\alpha}$-bound FP with PGE$_2$-bound EP2 and EP4 to compare their overall receptor conformations and ligand-binding pockets. A notable structural difference occurs in the H8 of these receptors. The H8 of FP is almost perpendicular to that of EP2, with a rotation of 82.3°, and is closer to the cell membrane compared with the H8 of EP4. (Fig. 1e). In

addition, although these ligands share a similar chemical scaffold and relatively conserved binding pocket in these three receptors, PGF$_{2\alpha}$ displays a distinct binding pose from PGE$_2$ in these complexes (Fig. 1e), as detailed below.

### The PGF$_{2\alpha}$ binding pocket of FP

The endogenous ligand PGF$_{2\alpha}$ is mainly composed of three parts, a carboxyl group-containing α-chain, a five-membered ring (F ring) with two hydroxyl groups, and a hydrophobic ω chain with one hydroxyl group at ω6 position (Fig. 2a). The cryo-EM map enabled the unambiguous assignment of PGF$_{2\alpha}$ within the receptor pocket. PGF$_{2\alpha}$ is well resolved in the FP ligand-binding pocket by adopting an L-shape conformation with its carboxyl group-containing α-chain fitting into a hydrophilic sub-pocket near the top of the receptor (Fig. 2a), formed by residues from TM1, TM7, and ECL2 (Fig. 2b). The carboxyl group in the α-chain of PGF$_{2\alpha}$ forms a salt bridge with R291$^{7.40}$ and hydrogen bonds with T184$^{EL2}$ and Y92$^{2.65}$. Mutations of residues R291$^{7.40}$, T184$^{EL2}$, and Y92$^{2.65}$, which are highly conserved in prostanoid receptors and participate directly in receptor binding, lead to decreased activity of PGF$_{2\alpha}$ (Fig. 2b–d, Supplementary Fig. 5-6). Although electrostatic contacts are the major driving force for the interactions between the α-chain and the positively charged binding pocket of FP, several hydrophobic residues also play important roles. Particularly, we did see the side chain of M115$^{3.23}$ forms lone pair-π interaction with ethylene linkage in the α-chain (Fig. 2b, c, Supplementary Fig. 6). Mutating this methionine in FP to alanine decreased the affinity for PGF$_{2\alpha}$ by approximately 100-fold (Fig. 2d, Supplementary Fig. 5). The F ring is located in a sub-pocket formed by TM1, TM2, and TM7 (Fig. 2b). These two hydroxyl groups in F ring mainly participate in polar interactions with the receptor. Particularly, these hydroxyl groups form hydrogen bonds with S33$^{1.39}$ and T294$^{7.43}$, which are not conserved among prostanoid receptor family members, indicating that this part mainly contributes to the selectivity of PGF$_{2\alpha}$ to FP. Meanwhile, mutation of S33$^{1.39}$ or T294$^{7.43}$ in FP to alanine significantly impaired the affinity to PGF$_{2\alpha}$ (Fig. 2d, Supplementary Fig. 5). Besides α chain and F ring, the ω chain penetrates into the hydrophobic pocket formed by TM5, TM6, and TM7. This alkyl chain forms hydrophobic interactions with F205$^{5.41}$, W262$^{6.48}$, F265$^{6.51}$, and L290$^{7.39}$. Mutations of these residues in FP to alanine significantly impaired the affinity to PGF$_{2\alpha}$ (Fig. 2d, Supplementary Fig. 5). Besides hydrophobic interaction, the ω6 hydroxy group forms a hydrogen bond with H81$^{2.54}$, which mutated to alanine also caused a dramatically reduced the activity of PGF$_{2\alpha}$ to FP by over 1000 folds (Fig. 2d, Supplementary Fig. 5). Collectively, the α-chain and ω-chain that bind to the sub-pockets with highly conserved residues, mainly contributes to high receptor binding affinity, while the F ring could be important for receptor selectivity.

### Specific engagement of LTPA and TFPA with FP

Even though FP is an important therapeutic target for many diseases, the poor selectivity of PGF$_{2\alpha}$ has hampered its clinical application. LTPA and TFPA, two synthetic relative selective FP agonists, have been widely used in clinical treatment for diseases including glaucoma. LTPA showed high affinity to FP with an EC$_{50}$ of 3.6 nM but only moderate potency to EP1 and EP3, with an EC$_{50}$ of 6.9 and 17 µM, respectively[27]. Compared to PGF$_{2\alpha}$, the selectivity was enhanced by more than 100 times[27]. TFPA is the most potent FP agonist (EC$_{50}$: 0.4 nM) and has a fairly low potency to the other members of the prostanoid receptor family except for EP3 (EC$_{50}$: 67 nM)[28]. LTPA, TFPA, and PGF$_{2\alpha}$ all have shared α-chain and F ring, while LTPA and TFPA have bulky substitutions in the ω-chain compared to PGF$_{2\alpha}$ (Fig. 3 a-e). LTPA and TFPA bond to FP in nearly identical binding poses as PGF$_{2\alpha}$ did, displaying a similar "L"-shape configuration in these two solved structures (Fig. 3a). Likewise, the carboxylate groups of LTPA and TFPA form strong polar interactions with R291$^{7.40}$, T184$^{EL2}$, and Y92$^{2.65}$, the

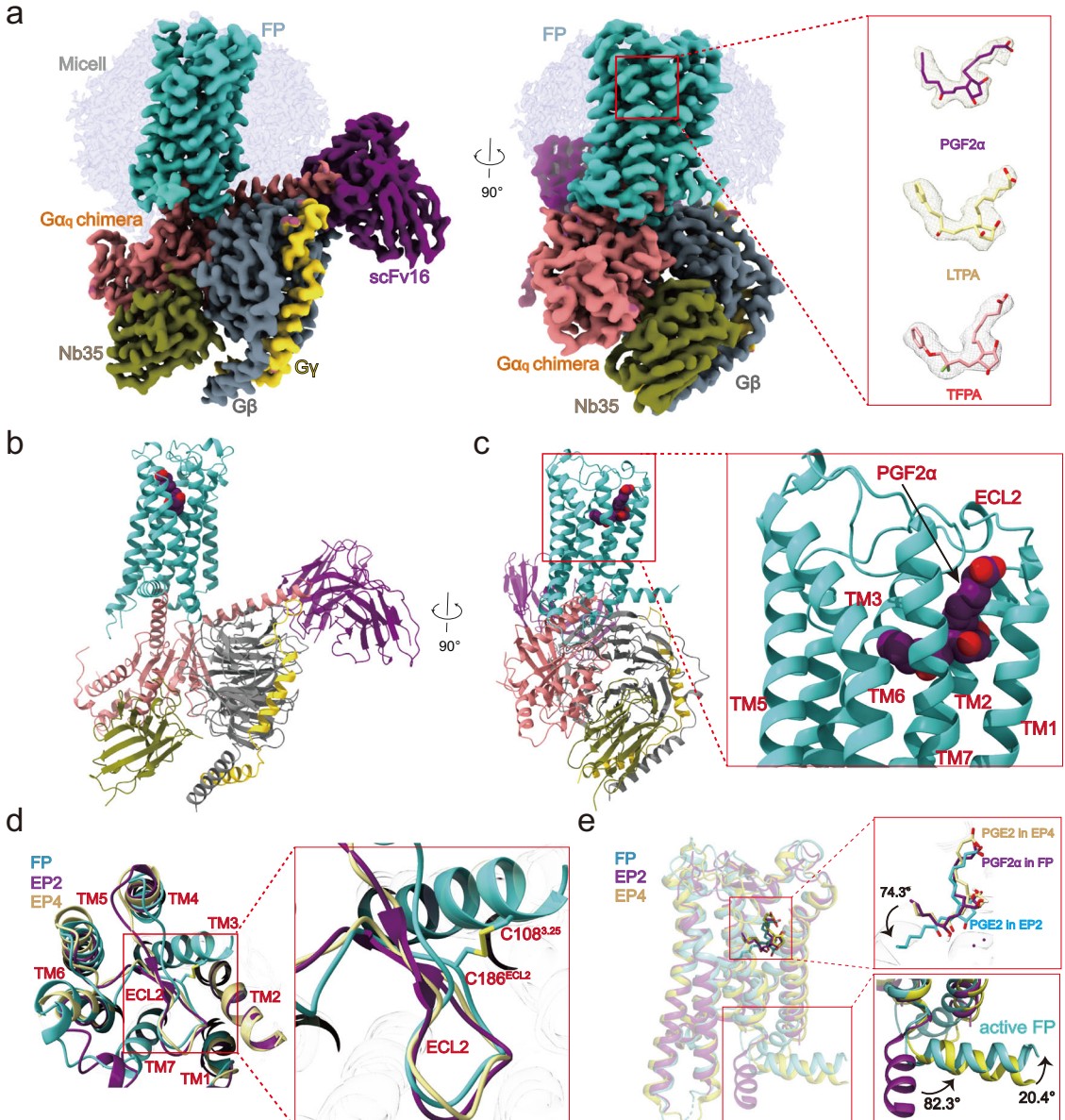

**Fig. 1 | Cryo-EM structures of FP−G$_q$ complexes. a** Cryo-EM density of FP-G$_q$ in complex with PGF$_{2\alpha}$, LTPA, or TFPA. FP in Medium Turquoise, Gα$_q$ in light Coral, Gβ in slate gray, Gγ in gold, ScFv16 in dark magenta, Nb35 in Olive, PGF$_{2\alpha}$ in purple, LTPA in yellow, and TFPA in light coral. **b, c** Cartoon representation of the PGF$_{2\alpha}$-FP-G$_q$ complex. FP in Green, Gα$_q$ in brown, Gβ in slate gray, Gγ in gold, ScFv16 in dark magenta, Nb35 in Olive, and PGF$_{2\alpha}$ in purple. **d, e** Comparison of the FP−G$_q$ complex with the EP2−Gs complex (PDB ID: 7CX2) and the EP4−Gs structure (PDB ID: 7D7M). FP shows a conserved conformation of ECL2 (green) with the other two receptors. The orientation of H8 of FP is different from the Gs-coupled receptor EP2 and EP3, and the ligand binding pose PGF$_{2\alpha}$ to FP is different from PGE$_2$ to EP2.

ethylene linkage in the α-chain forms a lone pair-π[29] interaction with M115[3.23], and the two hydroxy groups in the F ring form hydrogen bonds with S33[1.39] and T294[7.43] (Fig. 3b, c). Consistent with this observation, mutations of these residues in FP to alanine significantly decreased the potency of LTPA and TFPA to FP.

The derivatized ω-chain in LTPA also directly binds to FP. The phenyl group in the ω-chain of LTPA packs against F205[5.41], F265[6.51], F187[ECL2], L290[7.39], and W262[6.48] with hydrophobic interactions. These residues are highly conserved in FP, EP1, and EP3. Interestingly, besides forming a hydrogen bond with H81[2.54], the hydroxyl group in the ω chain forms an additional hydrogen bond with S118[3.35], which only exists in FP (Fig. 3d−f), indicating that this residue is of vital importance for the selectivity of LTPA to FP. Consistent with this finding, mutation of this serine to alanine or asparagine significantly reduced the potency of LTPA to induce FP activation (Fig. 3g, Supplementary Fig. 5).

Similar to LTPA, the phenyl group in the ω-chain of TFPA forms extensive hydrophobic interactions with FP. Moreover, the oxygen atom forms a hydron bond with the side chain of Q297, and the two fluoride groups in the carbon 12-position of TFPA form three hydrogen bonds with several residues in FP[30], which may explain its high potency to FP. One fluoride group in TFPA forms hydrogen bonds to H81[2.54] and S118[3.35], as the hydroxyl group in LTPA did. Notably, the other fluoride group forms a hydrogen bond with N84[2.57] (Fig. 3e, Supplementary Fig. 6). This residue is highly diverse among prostanoid receptor family members. Intriguingly, among the other 8 prostanoid receptors, only EP3 and TP harbor a similar polar uncharged residue, threonine (Fig. 3f). Through structural analysis, mutation of N84[2.57] to threonine in FP could still form a hydrogen bond with the fluoride group (Supplementary Fig. 6). This explains its strong affinity for EP3. Consistent with this prediction, mutation of this residue to threonine does not significantly reduce ligand binding of TFPA, but mutating to alanine

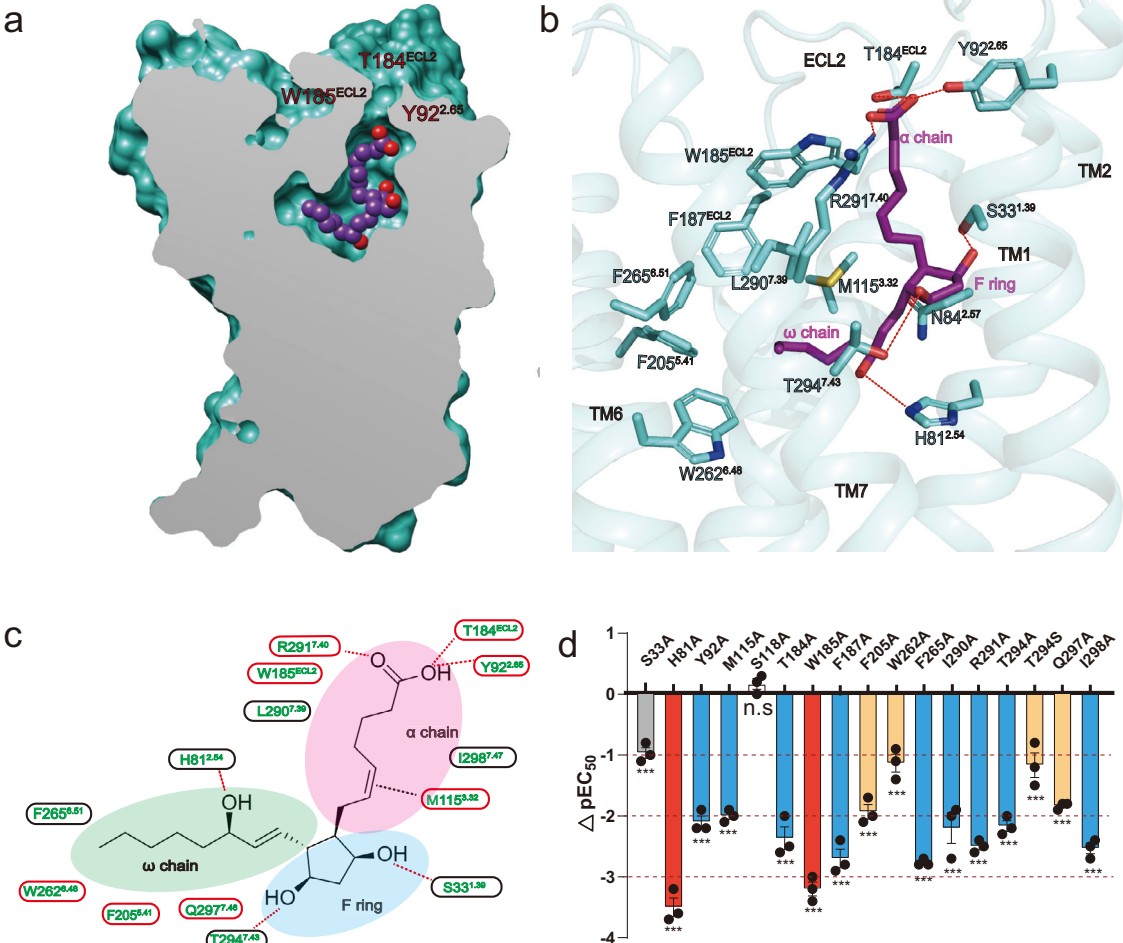

**Fig. 2 | The PGF$_{2\alpha}$ binding pocket of FP. a** Vertical cross-section of the PGF$_{2\alpha}$-binding pocket in FP. **b** Corresponding interactions that contribute to PGF$_{2\alpha}$ binding in FP. The hydrogen bond is depicted as a red dashed line. **c** Schematic representation of the interactions between FP and PGF$_{2\alpha}$ in 2D format and region division in FP–PGF$_{2\alpha}$ binding pockets corresponding to the structure of PGF2$\alpha$ (F ring, $\omega$ chain, and $\alpha$ chain). **d** IP1 accumulation assay of key mutants in FP that bind to PGF$_{2\alpha}$ ($\Delta$pEC$_{50}$ = pEC$_{50}$ of PGF$_{2\alpha}$ to specific Mutant-pEC$_{50}$ of PGF$_{2\alpha}$ to WT, Yellow column means $\Delta$pEC$_{50} \leq -1$, Blue column means $\Delta$pEC$_{50} \leq -2$, Red column means $\Delta$pEC$_{50} \leq -3$). Data are presented as mean values $\pm$ SEM; $n = 3$ independent samples; significance was determined with a two-sided unpaired $t$-test; n.s. no significant; $^*p < 0.05$; $^{**}p < 0.01$; $^{***}p < 0.001$. Exact $p$ values and Source data are provided as a Source Data file.

does (Fig. 3g, Supplementary Fig. 5). This hydrogen bond formed between T$^{2.57}$ in EP3 and TFPA may contribute to the high affinity of TFPA toward EP3[28].

**The active structure of FP**

Structural comparison of the PGF$_{2\alpha}$-bound FP-G$_q$ complexes with the antagonist-bound TP (PDB: 6IIU)[31] supports the notion that FP in these structures is in the active state, featured by the outward displacement of the cytoplasmic end of TM6, the hallmark of class A GPCR activation, and concurrently inward shift of TM7. In addition, TM5 of FP laterally shifts relative to that of antagonist-bound TP. These conformation changes largely resemble that of the G$_q$-coupled 5-HT$_{2A}$R complex (PDB: 6WHA)[22] (Fig. 4a), but the outward amplitude of TM6 of FP is smaller than that of 5-HT$_{2A}$R (Fig. 4a).

Structure comparison of the active FP in complex with PGF$_{2\alpha}$ and the inactive TP bound to an antagonist provide clues for understanding the activation mechanism of FP. Upon PGF$_{2\alpha}$ binding, the $\omega$-chain of PGF$_{2\alpha}$ approaches the toggle switch residue W262$^{6.48}$ to trigger the downward displacement of W262$^{6.48}$ by 2.4 Å (Fig. 4b, c). The movement of W262$^{6.48}$ further constitutes a hydrophobic LLW core triad, which is comprised of L123$^{3.40}$, L213$^{5.49}$, and W262$^{6.48}$, to fasten TM3, TM5, and TM6. The importance of the LLW core triad is

functionally supported by the decreased potency of PGF$_{2\alpha}$ to activate L123$^{3.40}$A and L213$^{5.49}$A FP mutants (Fig. 4c, Supplementary Fig. 7). Notably, the canonical P$^{5.50}$ I$^{3.40}$ F$^{6.44}$ motif in majority of class A GPCRs[26,32] is replaced by G214$^{5.50}$ L123$^{3.40}$ S258$^{6.44}$. In our structure of active FP, G214$^{5.50}$ is sterically apart from S258$^{6.44}$ and L123$^{3.40}$, leading to the lack of the conserved PIF hydrophobic triad, thus indicating that FP employs an activation mechanism not mediated by the traditional PIF motif (Fig. 4b).

The receptor activation also accompanies the arrangement of ionic lock (D/E$^{3.49}$ R$^{3.50}$ Y$^{3.51}$, E132$^{3.49}$ R133$^{3.50}$ C134$^{3.51}$ in FP), leading to the broken of the salt bridge between E132$^{3.49}$ and R133$^{3.50}$ and the stretching of the R133$^{3.50}$ side chain towards TM6 and the latter's outward displacement of 6.7 Å compared with that of inactive TP (measured at C$\alpha$ of residue 6.30). C134$^{3.51}$ in the ERC motif moves inwards, forming interactions with I222$^{5.58}$ and T223$^{5.59}$ in TM5. The importance of motif in FP activation is evidenced by the decreased PGF$_{2\alpha}$ activity on FP mutants of E132$^{3.49}$, R133$^{3.50}$, and C134$^{3.51}$ (Supplementary Fig. 7). Intriguingly, unlike its cognate residue R312$^{8.47}$ in TP, the side chain of R308$^{8.47}$ in FP undergoes a large-scale upward rotation and forms a hydrogen bond with Q250$^{6.36}$, which leads to the inward shift of the cytoplasmic end of TM7 and the upshift of H8 (Fig. 4d). The importance of this hydrogen bond in FP activation is evidenced by the loss of

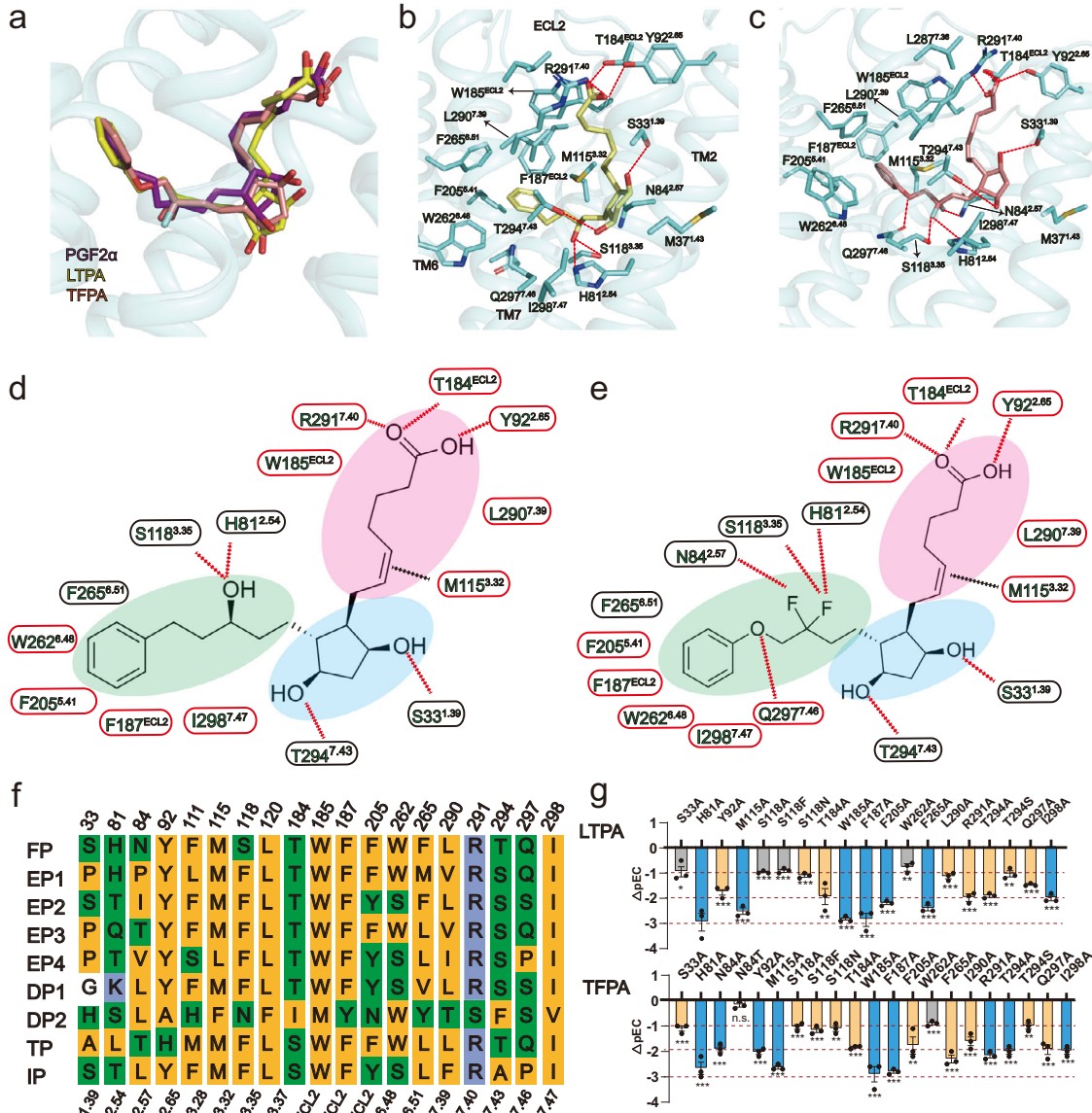

**Fig. 3 | Specific engagement of LTPA and TFPA with FP. a** Superimposition of the PGF$_{2\alpha}$ bound FP with LTPA bound FP and TFPA bound FP aligned at the ligand binding pocket. **b**, **c** Detail interactions of LTPA-FP and TFPA-FP were shown. H-bonds were depicted as red dashed lines. **d**, **e** 2D representation of contacts in LTPA-FP and TFPA-FP. The polar bonds were presented by red dotted lines. Conserved residues that formed interactions with ligands are presented in a red oval frame. **f** Sequence alignment of prostanoid receptors. Hydrophobic residues are in yellow, polar charged residues in blue, and polar uncharged residues in green. **g**, IP1 accumulation assay of key mutants in FP that bind to LTPA or TFPA ($\Delta pEC_{50} = pEC_{50}$ of agonists to specific Mutant FP-$pEC_{50}$ of PGF$_{2\alpha}$ to WT FP, Yellow column means $\Delta pEC_{50} \leq -1$, Blue column means $\Delta pEC_{50} \leq -2$). Data are presented as mean values ± SEM; $n = 3$ independent samples; significance was determined with two-side unpaired $t$-test; n.s. no significant; *$p < 0.05$; **$p < 0.01$; ***$p < 0.001$. Exact $p$ values and Source data are provided as a Source Data file.

PGF$_{2\alpha}$ activity on FP mutants of R133$^{3.50}$A, Q250$^{6.36}$A, and R308$^{8.47}$A. (Supplementary Fig. 7) Noteworthily, the arginine at position 8.47 is highly conserved across prostaglandin receptors, while Q$^{6.36}$ only exists in several class A receptors, including EP1, EP3, and FP, indicating that this interaction network could also exist in active structures of EP1 and EP3.

## FP-G$_q$ coupling

The notable outward displacement of TM6 at the cytoplasmic side opens a cavity to accommodate the G$\alpha_q$ subunit. Structural comparisons of FP-G$_q$ with G$_q$-coupled CCK$_A$R and 5HT$_{2A}$R reveal a difference in the conformations of TM1, TM6, and G$\alpha_q$ subunits among these G$_q$-coupled GPCR complexes (Fig. 5a). The cytoplasmic end of FP TM1 and TM6 undergoes a remarkably inward displacement relative to CCK$_A$R and 5HT$_{2A}$R. Consequently, the C-terminus of the $\alpha$5 helix of the G$\alpha_q$ subunit in the FP-G$_q$ complex rotates toward TM7 and H8 to avoid clashes with TM6 and forms extra hydrophobic interactions with side chains of residues in H8, accompanied by the rotation of the entire G$\alpha_q$ subunit (Fig. 5a). This alteration triggers a 14° tilt of the $\alpha$N helix of G$\alpha_q$, bringing it closer to the cytoplasmic end of TM4 in the FP-G$_q$ complex relative to CCK$_A$R-G$_q$ and 5HT$_{2A}$R-G$_q$ complexes (Fig. 5a). In addition to coupling G$_q$ by interactions with H8 and TM4, the engagement of G$_q$ is also maintained by interactions with FP from TM1, TM2, TM3, TM5, TM4, TM6, ICL1, and ICL2. The side chain of E132$^{3.49}$ of the E$^{3.49}$ (D) R$^{3.50}$C$^{3.51}$ (Y) motif in TM3 makes a direct hydrogen bond interaction with the side chain of the Y356 of $\alpha$5 (Fig. 5b, c).

Like other G$_q$-coupled GPCR[33], the ICL2 of FP facilitates broad interactions with G$_q$ to stabilize the complex. Typically, ICL2 adopts a helix conformation and inserts into the groove formed by the $\alpha$N, $\beta$2–$\beta$3 loop, and $\alpha$5 helix of G$\alpha_q$. Interestingly, we identified several distinct interactions between TM6 and G$\alpha_q$. For instance, the H244$^{6.30}$ forms a hydrogen bond interaction with the side chain of Q350 of G$\alpha_q$,

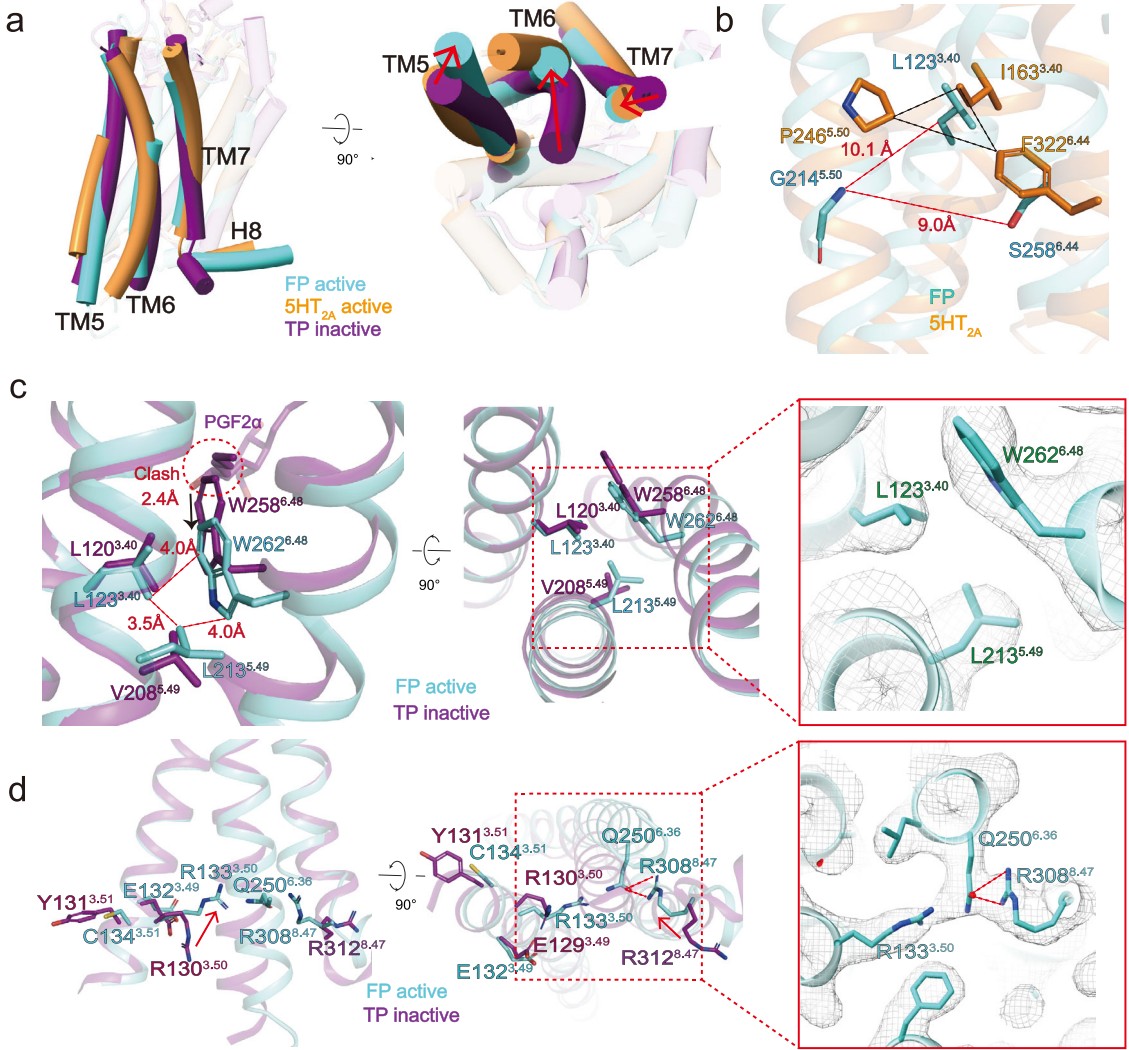

**Fig. 4 | The active structure of the FP. a** Superimposition of $G_q$-coupled FP with the $G_q$-coupled 5-HT$_{2A}$R complex and antagonist-bound TP (PDB: 6IIU). FP in Green, 5-HT$_{2A}$R in brown, and TP in purple. **b** Superimposition of $G_q$-coupled FP with $G_q$-coupled 5-HT$_{2A}$ aligned at the PIF motif. **c** Superimposition of $G_q$-coupled FP with antagonist-bound TP aligned at the LLW motif. The left panel is a magnified view of the LLW motif. The right panel is the top view of the LLW motif. Residues are shown in the sticks, with the correspondent cryo-EM density represented in the mesh. **d** Superimposition of $G_q$-coupled FP with antagonist-bound TP aligned at the D(E)RY(C) motif and RQR motif. The left panel is a magnified view of the D(E)RY(C) motif and RQR motif. The right panel is the top view of D(E)RY(C) and RQR motifs. Residues are shown in the sticks, with the correspondent cryo-EM density represented in the mesh.

and E246$^{6.32}$ forms a hydrogen bond interaction with the side chain of N357 of G$\alpha_q$ (Fig. 5d). Mutating of H244$^{6.30}$ and E246$^{6.32}$ to alanine destabilizes the complex and reduces the IP1 accumulation of FP (Fig. 5d, Supplementary Fig. 6). Most notably, unlike other solved G$_q$-coupled GPCR structures, the extensive interactions of TM1, ICL1, and TM2 of FP with G$_q$ play an important role in stabilizing the complex (Fig. 5e). F58 packs against L358 and V359 of G$\alpha_q$ stabilized by hydrophobic interaction, and K63 packs against R37 of G$\alpha_q$. S62 forms a hydrogen bond with E355 of G$\alpha_q$ (Fig. 5e). Mutating S62 to alanine almost abolished the IP1 accumulation of FP (Fig. 5e, Supplementary Fig. 7), supporting that the direct interactions of TM1, ICL1, and TM2 with G protein are important for FP to couple with G protein.

## Discussion
Glaucoma is the leading cause of permanent eyesight loss in the world[34]. According to the World Health Organization (WHO), 79.6 million individuals suffered from glaucoma in 2020, and the number of patients may increase to 111.8 million by 2040[35]. Activation of prostaglandin receptors like FP, EP1–EP4, and DP showed anti-glaucoma effects. Research for the discovery of pharmaceutical drugs selectively targeting each of these receptors has been extensively conducted,

whereas only FP selective agonists are approved for treatment due to their minimal side-effect profile[36]. Elucidating the mechanism underlying the functional selectivity of individual FP receptors is of utmost importance for developing new drugs with higher selectivity towards individual prostaglandin receptors to avoid or reduce undesirable side effects.

In this study, we present the cryo-EM structures of FP-G$_q$ in complex with its endogenous ligand PGF$_{2\alpha}$ as well as two synthetic agonists LTPA and TFPA. Through structure analysis of the PGF$_{2\alpha}$–FP–G$_q$ complex, the carboxylate-containing α-chain that binds to the sub-pocket with highly conserved residues contributes to the majority of the high potency of the receptor. In contrast, the F ring is important for receptor selectivity. Similar to PGE$_2$ to EP3, the hydrophobic ω-chain directly interacts with the toggle switch W262$^{6.48}$ in FP and forms hydrophobic interactions with a set of hydrophobic residues. The hydroxyl group in the ω-chain of LTPA forms an additional hydrogen bond with S118$^{3.35}$ in FP in the LTPA-FP-G$_q$ structure, which may be the key to LTPA's higher selectivity to FP. Despite PGF$_{2\alpha}$ containing the hydroxyl group at the same position as LTPA that the presence of the aromatic ring and lack of carbon-to-carbon double bonds in the ω-chain of LTPA may affect the geometry of the ligand,

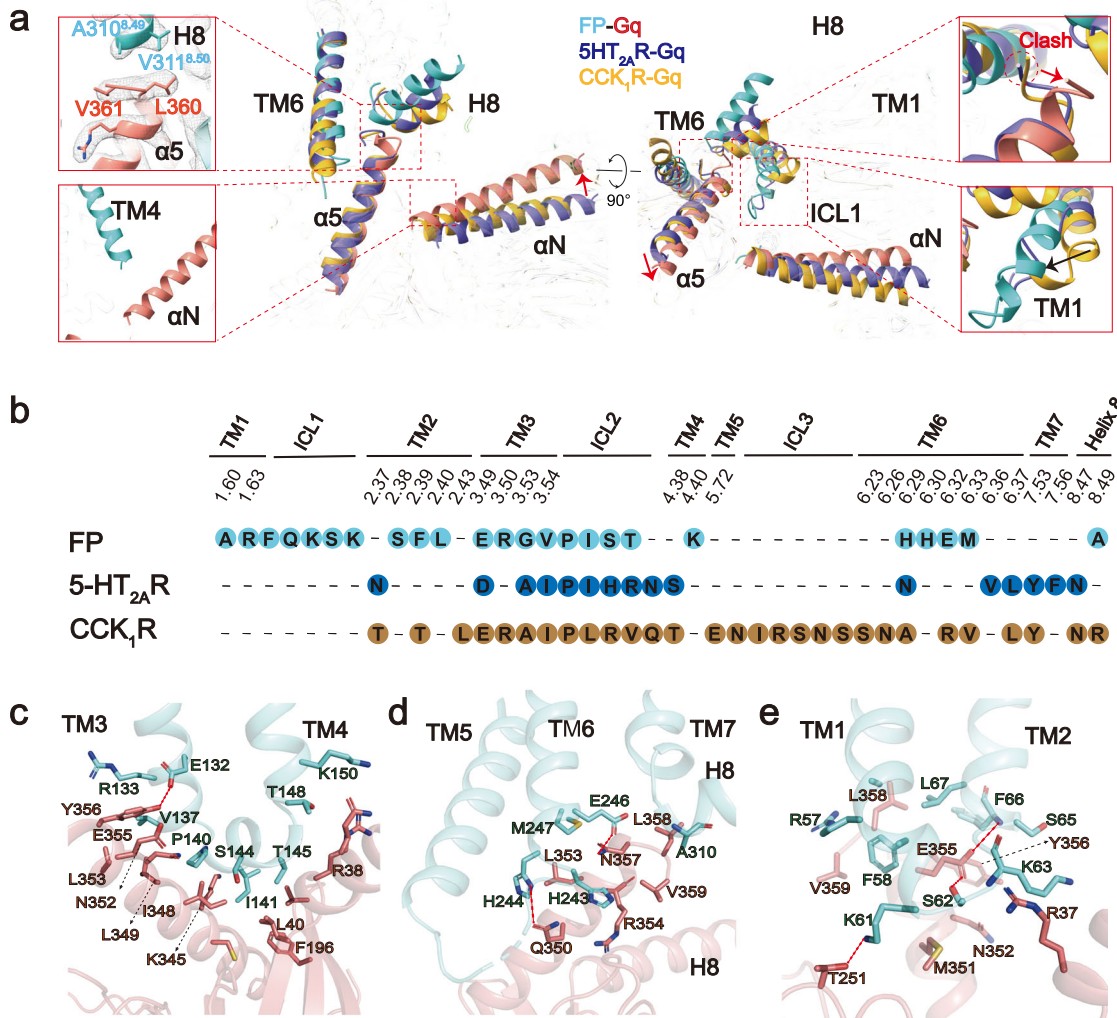

**Fig. 5 | FP–G$_q$ coupling. a** The structures of G$_q$-coupled FP, 5-HT$_{2A}$R (PDB ID: 6WHY), and CCK$_1$R (PDB ID: 7MBY) complexes were superimposed based on TM2, TM3 and TM4, FP is shown in light sea green, 5-HT$_{2A}$R in blue and CCK$_1$R in yellow. This panel is shown with views an orthogonal view (left) and a cytoplasmic view (right); the red arrows indicate the tilt of the α5 helix of Gα$_q$ from the FP–G$_q$ complex compared to the 5-HT$_{2A}$R–G$_q$ or CCK$_{A1}$–G$_q$ complexes. **b** the residues in FP, 5-HT$_{2A}$R, and CCK$_{A1}$ that contact G$_q$. **c–e** The detailed interactions of ICL2 with Gα$_q$ (**c**), TM6 with the α5 helix of Gα$_q$ (**d**) and TM1, TM2, and ICL1 with the αN and α5 helices of Gα$_q$ (**e**). The hydrogen bonds are depicted as dashed lines.

indicating that these modifications in the ω-chain of PGAs may improve their selectivity to FP. Compared with the LTPA, TFPA forms an extra hydrogen bond to N84$^{2.57}$ in FP with a fluoride group in the ω chain. EP3 harbors a similar polar uncharged residue, T$^{2.57}$, which may form a hydrogen bond with the fluoride group of TFPA. Although TFPA had a higher affinity for FP than LTPA, its selectivity needs to be further improved. These structures revealed the ligand recognition of FP, which will serve as templates for the rational design of a new generation of potent agonists with desired selectivity profiles.

Through structural comparison and mutagenesis studies, we also elucidated the mechanisms of receptor activation and G protein coupling by FP. FP has neither the traditional 'PIF' core triad nor the DRY motif that is commonly involved in the activation of class A GPCRs, instead FP senses ligand binding by the conserved toggle switch W$^{6.48}$ to tether TM3–TM5–TM6 through a hereby identified LLW core triad consisting of L123$^{3.40}$, L213$^{5.49}$. Intriguingly, mutation of the W262$^{6.48}$ in FP to alanine did not entirely abolish these three agonists induced activation of FP. It is speculated that the ω chain of PGF$_{2α}$, LTPA, or TFPA in W262A mutant still induces the activation of mutated FP because it may point downward in a similar manner to EP2 (Fig. 1e), which naturally lacks W$^{6.48}$ at the toggle switch residue position[37]. In the FP receptor, the conserved D/ERY motif is replaced

by ERC, which only exists in several class A GPCRs, like EP1, IP, and Neurotensin receptor type 2. Interestingly, in the active FP structure, Q250$^{6.36}$ induces an upswing of the side chain of R308$^{8.47}$ and forms a hydrogen bond with it that leads to the inward shift of the cytoplasmic end of TM7 and the upshift of H8. R$^{8.47}$ is particularly conserved in prostaglandin receptors but rarely exists in other class A GPCRs, and Q$^{6.36}$ only exists in several class A receptors, including EP1, EP3, and FP. All these key residues for activation in FP are harbored by EP1, which also primarily couples with G$_q$. Thus, this activation mechanism should be applicable to both FP and EP1, which have the same unique residues.

The distinct features of the active FP structures also define the way of G$_q$ protein coupling. Specifically, the TM1, ICL1, and TM2 of FP form extensive interactions with G$_q$, which, to our knowledge, haven't been shown in any other reported G$_q$-coupled GPCR structures. These observed characteristics in the FP-G$_q$ complex structures, including the arrangement of the 7TM bundle, the ligand-binding mode, the ligand-induced receptor activation, and the manner of G protein coupling, expand the understanding of lipid recognition and GPCR-G$_q$ coupling mechanism. Collectively, our results reveal conserved and diverse mechanisms of ligand binding, receptor activation, and G protein coupling by FP.

# Methods

## Constructs

The full-length human FP was modified to contain the N-terminal thermally stabilized BRIL[20] to enhance receptor expression and the addition of an N-terminal Flag tag. LgBiT was inserted at the C-terminus of the human FP using homologous recombination. The modified FP was cloned into the pFastBac (Thermo Fisher Scientific) vectors using the ClonExpress II One Step Cloning Kit (Vazyme Biotech). An engineered $G\alpha_q$ chimera was generated based on the mini-$G\alpha_s$ scaffold with its aa1-18 replaced by corresponding sequences of $G\alpha_{i1}$, aa348-359 replaced by corresponding sequences of $G\alpha_q$, A33, H35, A87, V92, D107, V115, R137, N144, C151, F155, K158, V161, K163, D171, and D319 replaced by corresponding residues in $G\alpha_q$, designated as $mG\alpha_{s/q/iN}$. Human wild-type (WT) $G\beta1$, human $G\gamma2$, and a single-chain antibody scFv16[38], as well as a $G\beta1$ fused with SmBiT at its C-terminus, were cloned into pFastBac vectors.

## Insect cell expression

Human FP, $G_q$ chimera, $G\beta1$, $G\gamma$, and scFv16 were co-expressed in High Five insect cells (Invitrogen) using the baculovirus method (Expression Systems). Cell cultures were grown in ESF 921serum-free medium (Expression Systems) to a density of 2–3 million cells per mL and then infected with six separate baculoviruses at a suitable ratio. The culture was collected by centrifugation 48 h after infection, and cell pellets were stored at −80 °C.

## Complex purification

Cell pellets were thawed in 20 mM HEPES pH 7.4, 150 mM NaCl, 10 mM $MgCl_2$, and $CaCl_2$ supplemented with Protease Inhibitor Cocktail (TargetMol). For the $PGF_{2\alpha}$/LTPA/TFPA-FP-$G_q$-scFv16 complexes, 10 μM $PGF_{2\alpha}$/LTPA/TFPA (MedChemExpress) and 2 mg Nb35 were added. The suspension was incubated for 1 h at room temperature, and the complex was solubilized from the membrane using 0.5% (w/v) lauryl maltose neopentyl glycol (LMNG) (Anatrace) and 0.1% (w/v) cholesteryl hemisuccinate (CHS) (Anatrace) for 2 h at 4 °C. Insoluble material was removed by centrifugation at 70,000$g$ for 35 min, and the supernatant was immobilized on the Flag resin (SinoBiological). The resin was then packed and washed with 30 column volumes of 20 mM HEPES pH 7.4, 150 mM NaCl, 0.01% (w/v) LMNG, 0.002% CHS, and 10 μM ligand. The complex sample was eluted in buffer containing 20 mM HEPES pH 7.4, 150 mM NaCl, 0.01% (w/v) LMNG, 0.002% CHS, 10 μM ligand, and 0.2 mg/ml FLAG peptide (GenScript). Complex fractions were concentrated with a 100-kDa molecular weight cut-off (MWCO) Millipore concentrator for further purification. The complex was then subjected to size-exclusion chromatography on a Superdex 6 Increase 10/300 GL column (GE Healthcare) pre-equilibrated with size buffer containing 20 mM HEPES pH 7.4,150 mM NaCl, 0.00075% (w/v) LMNG, 0.00025% (w/v) GDN (Anatrace), 0.00025% digitonin (w/v), 0.00015% CHS, and10 μM ligand to separate complexes. Eluted fractions were evaluated by SDS-PAGE, and those consisting of receptor-$G_q$ protein complexes were pooled and concentrated for cryo-EM experiments.

## Cryo-EM data collection

Cryo-EM grids were prepared with the Vitrobot Mark IV plunger (FEI) set to 4 °C and 100% humidity. Three microliters of the sample were applied to the glow-discharged gold R1.2/1.3 holey carbon grids. The sample was incubated for 10 s on the grids before blotting for 3 s (double-sided, blot force −2) and flash-frozen in liquid ethane immediately. For FP−$G_q$−$PGF_{2\alpha}$ complex, FP-$G_q$-LTPA complex, and FP-$G_q$-TFPA complex datasets, 3902, 7972, and 6513 movies were collected, respectively, on a Titan Krios equipped with a Gatan K3 direct electron detection device at 300 kV with a magnification of 105,000, corresponding to a pixel size 0.824 Å. Image acquisition was performed with EPU Software (FEI Eindhoven, Netherlands). We collected a total of 36 frames accumulating to a total dose of 50 e⁻ Å⁻² over 2.5 s exposure.

## Cryo-EM image processing

MotionCor2 was used to perform the frame-based motion-correction algorithm to generate a drift-corrected micrograph for further processing, and CTFFIND4 provided the estimation of the contrast transfer function (CTF) parameters[39,40]. All subsequent steps, including particle picking and extraction, two-dimensional (2D) classification, three-dimensional (3D) classification, 3D refinement, CTF refinement, Bayesian polishing, post-processing, and local resolution estimation, were performed using Relion3.0[41].

For FP−$G_q$−$PGF_{2\alpha}$ complex dataset, 115 aligned micrographs were deleted because of contaminations or bad ice quality. A total of 3,391,620 particles were extracted from the cryo-EM micrographs and followed by two rounds of reference-free 2D classification, yielding 895,825 particles after clearance. Mask 3D classification on the receptor part was used to separate out 479,164 particles which resulted in a clearer density of PTGFR. We refined these particles, which led to a structure at 3.24 Å global resolution. After the postprocessing, the particles were reconstituted to a 2.67 Å structure (Supplementary Fig. 1).

For FP−$G_q$−LTPA complex dataset, 1438 aligned micrographs were deleted because of contaminations or bad ice quality. A total of 5,478,774 particles were extracted from the cryo-EM micrographs and followed by two rounds of reference-free 2D classification, yielding 1,181,590 particles after clearance. The global 3D classification was used to separate out 437,740 particles. We then continued the processing in Relion3.0 and refined 437,740 particles, which led to a structure at 3.40 Å global resolution. After CTF refinement, Bayesian polishing, and postprocessing, then the particles were reconstituted to a 2.78 Å structure (Supplementary Fig. 2).

For FP−$G_q$−TFPA complex dataset, 279 aligned micrographs were deleted because of contaminations or bad ice quality. A total of 5,774,308 particles were extracted from the cryo-EM micrographs and followed by two rounds of reference-free 2D classification, yielding 2,418,674 particles after clearance. The global 3D classification was used to separate out 805,802 particles. Two rounds of mask 3D classification on the receptor part were used to separate out 578,962 particles which resulted in a clearer density of PTGFR. We refined these particles, which led to a structure at 3.35 Å global resolution. After CTF refinement, Bayesian polishing, and postprocessing, then the particles were reconstituted to a 3.14 Å structure (Supplementary Fig. 3). We also performed postprocessing of all three final maps with DeepEMhancer[42].

## Model building

A predicted FP structure from Alphafold2 was used as the starting reference model for receptor building[43]. Structures of $G_{\alpha q}$, $G\beta$, $G\gamma$, and the scFv16 were derived from PDB entry 7WKD[44] and were rigid body fit into the density. All models were fitted into the EM density map using UCSF Chimera[45], followed by iterative rounds of manual adjustment and automated rebuilding in COOT[46] and PHENIX[47], respectively. The model was finalized by rebuilding in ISOLDE[48], followed by refinement in PHENIX with torsion-angle restraints to the input model. The final model statistics were validated using Comprehensive validation (cryo-EM) in PHENIX[47] and provided in Supplementary Table 1. All structural figures were prepared using Chimera[45], Chimera X[49], and PyMOL (Schrödinger, LLC.).

## Inositol phosphate accumulation assay

IP-One production was measured using the IP-One HTRF kit (Cisbio)[50]. Briefly, AD293 cells (Agilent) were grown to a density of 400,000–500,000 cells per mL and then infected with separate plasmids at a suitable concentration. The culture was collected by

centrifugation 24 h after incubation at 37 °C in 5% $CO_2$ with a Stimulation Buffer. The cell suspension was then dispensed in a white 384-well plate at a volume of 7 µl per well before adding 7 µl of ligands. The mixture was incubated for 1 h at 37 °C. IP-One-d2 and anti-IP-One Cryptate dissolved in Lysis Buffer (3 µl each) were subsequently added and incubated for 15-30 min at room temperature before measurement. Intracellular IP-One measurement was carried out with the IP-One HTRF kit and EnVision multi-plate reader (PerkinElmer) according to the manufacturer's instructions. Data were normalized to the baseline response of the ligand. $pEC_{50}$ $E_{min}$, and $E_{max}$ for each curve were calculated by GraphPad Prism 8.0. $\Delta pEC_{50}$ equals $pEC_{50}$ of agonists to specific Mutant minus $pEC_{50}$ of agonists to WT. Data are presented as mean values ± SEM; $n = 3$ independent samples; n.s. no significant; $*p < 0.05$; $**p < 0.01$; $***p < 0.001$.

### Receptor surface expression

Cell-surface expression levels of WT or mutants FP were quantified by flow cytometry. AD293 cells were seeded at a density of $1.5 \times 10^5$ per well into 12-well culture plates. Cells were grown overnight and then transfected with 1.0 µg FP construct by FuGENE® HD transfection reagent in each well for 24 h. After 24 h of transfection, cells were washed once with PBS and then detached with 0.2% (w/v) EDTA in PBS. Cells were blocked with PBS containing 5% (w/v) BSA for 15 min at room temperature before incubating with primary anti-Flag antibody (diluted with PBS containing 5% BSA at a ratio of 1:300, Sigma-Aldrich, F3165) for 1 h at room temperature. Cells were then washed three times with PBS containing 1% (w/v) BSA and then incubated with anti-mouse Alexa-488-conjugated secondary antibody (diluted at a ratio of 1:1000, Thermo Fisher, A-11029) at 4 °C in the dark for 1 h. After another three times of washing, cells were collected, and fluorescence intensity was quantified in a Luminex flow cytometer system (Guava® easyCyte) through a Luminex guavaSoft 4.5 at excitation 488 nm and emission 519 nm. Approximately 10,000 cellular events per sample were collected, and data were normalized to the wild-type FP. Experiments were performed at least three times, and data were presented as means ± SEM.

### Statistics

All functional study data were analyzed using GraphPad Prism 8.0 (Graphpad Software Inc.) and showed as means ± S.E.M. from at least three independent experiments in triplicate. The significance was determined with a two-sided, unpaired $t$-test, and $*p < 0.05$ was considered statistically significant.

### Reporting summary

Further information on research design is available in the Nature Portfolio Reporting Summary linked to this article.

## Data availability

The atomic coordinates and the electron microscopy maps have been deposited in the Protein Data Bank (PDB) under accession numbers 8IUK, 8IUL, and 8IUM and Electron Microscopy Data Bank (EMDB) accession number EMD-35724, EMD-35725, and EMD-35726 for the PGF2α–FP–$G_q$ and the LTPA–FP–$G_q$ and the TFPA–FP–$G_q$ complex, respectively. Previously published structures can be accessed via accession codes: 7CX2; 7D7M; 6IIU; 6WHY; 7MBY; 6WHA; 7WKD. Source data are provided in this paper.

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

## Acknowledgements

The cryo-EM data were collected at the Shanghai Advanced Center for Electron Microscopy, Shanghai Institute of Materia Medica, Chinese Academy of Sciences. We thank K.W., W.H., and Q.Y. for performing cryo-EM data collection. This work was partially supported by Ministry of Science and Technology (China) grants (2018YFA0507002 to H.E.X.); Shanghai Municipal Science and Technology Major Project (2019SHZDZX02 to H.E.X.); Shanghai Municipal Science and Technology Major Project (H.E.X.); CAS Strategic Priority Research Program (XDB37030103 to H.E.X.); The National Natural Science Foundation of China (32130022 to H.E.X., 82121005 to H.E.X., 32171187 to Y.J., 82121005 to Y.J.); China Postdoctoral Science Foundation Funded Project (2021M703342 to C.W.); Shanghai Post-doctoral Excellence Program (2021429 to C.W.); Key tasks of Lingang Laboratory (LG202101-01-03 to Y.X.); the National Natural Science Foundation of China (81902085 to Y.X.).

## Author contributions

C.W. designed the expression constructs and purified the protein complex supervised by H.E.X. Y.X. and J.D. prepared the grids. Y.X. performed cryo-EM data processing and model building. C.W. and Q.H. constructed all the mutated plasmids, and C.W. performed functional studies supervised by H.E.X. H.E.X. and C.W. analyzed the structures. C.W. prepared the figures and initial paper. C.Y. and Y.X. contributed to the preparation of the figures. Y.X. contributed to manuscript preparation. Q.H., D.L., H.C., C.L., and C.Y. helped with experiments. H.E.X. conceived the project and initiated collaborations with W.F., All authors discussed and commented on the paper. H.E.X. and Y.J. revised the paper, and H.E.X. supervised the project. H.E.X. and C.W. wrote the paper with input from all authors.

## Competing interests

H.E.X., Y.J., C.W., Y.X., Q.H., J.D., H.C., C.L., and C.Y. declare no competing interest. W.F. and D.L. are employees of Sironax (Beijing) Co., Ltd.
