## [Peer Review File · Nature Communications]

Ligand-induced activation and G protein coupling of prostaglandin F₂ α receptorEditorial Note: This manuscript has been previously reviewed at another journal that is not operating a transparent peer review scheme. This document only contains reviewer comments and rebuttal letters for versions considered at *Nature Communications*.

REVIEWERS' COMMENTS

Reviewer #1 (Remarks to the Author):

In this manuscript "Unique features of ligand-induced activation and G protein coupling of prostaglandin F2a receptor" by Canrong Wu, H. Eric Xu, and co-authors, they determined the structures of F2a receptor (FP) in complex with engineered Gq protein in the presence of three different agonists: endogenous ligand PGF2a, and synthetic ligand LTPA and TFPA. FP belongs to the prostaglandin receptor family of GPCRs playing pivotal roles in regulating inflammation, allergy, intraocular pressure, and blood pressure. Through the cryoEM structures of FP, the authors reveal the ligand binding mode of the receptor, unique signature motif associated activation, and the engagement mode with the downstream G protein by combining structural analysis, mutagenesis, and IP1 cellular signaling assay. Notably, the authors provide structural framework governing the ligand-receptor selectivity. Carboxyl group of the alpha-chain and hydrophobic nature of the omega-chain of the ligand are crucial for the tight receptor binding, and F-ring for the receptor specificity. The findings presented in this study are of particular importance for the design of the ligand that selectively modulates the target receptor among the prostaglandin family with reduced aversive effect.

The overall writing in this report is well described.

I have several comments and questions as listed below.

Having looked at the cryoEM density maps and the models, it seems that the density corresponding to the ligand is weak compared to the protein density surrounding the ligand for all three structures. At the low contour level, they all show up appropriately like Fig1a. This looks indicating low occupancy of the ligand. Given the high affinity of the ligands, it is hard to consider that the ligand falls off the pocket so easily. Do the authors have any comment?

Of the three ligands, PGF2a fails to form a hydrogen bond with S118. Close look at the omega-chain density of the ligand, I agree that the hydroxyl group of PGF2a points away from S118, while that of LTPA and fluoride of TFPA points toward S118. The authors briefly mention around this as potentially triggered by the presence of the aromatic ring. Given that the ligand selectivity is one of the main topics of this manuscript and S118 is unique to FP, this geometry switch should be discussed in more detail possibly even in the main text.

The authors claim that the FP possesses the unique activation mechanisms. Indeed, this receptor lacks both "PIF" and "E/DRY" motifs. However, the activation mechanism itself appears very similar to the classical mode: The agonist triggers a conformational change of the toggle switch "W6.48", which is transmitted to the "PIF" or an equivalent motif, then the "E/DRY" or an equivalent motif opens up to allow opening of TM6. It appears too much overemphasizing to claim FP activation undergoes unique mechanism including the title of the manuscript. I would rather prefer emphasizing more on the ligand selectivity from the structural analyses of these ligands.

The authors employed chimeric Gq protein for the structural study. It is described in the Methods section, but please clarify the boundary between Gs part and Gq part as well.

Line 247: "C1343.51 in the "ERC" motif moves inwards, forming a polar 247 interaction with T2235.59 in TM5" The geometry doesn't look these residues form a polar interaction.

Supplemental Figure 7a: The colors of the label don't match those of the curve.

Reviewer #2 (Remarks to the Author):

Canrong Wu and colleagues have provided a revised version of the manuscript, "Unique features of ligand-induced activation and G protein coupling of prostaglandin F2a receptor". The authors have appropriately addressed previous concerns and comments in a manner that is suitable for publication. The overall findings are well supported and described and will be of broad interest to the field.

Manuscript ID: Nature Communications Manuscript # NCOMMS-22-47949A

Title: Unique features of ligand-induced activation and G protein coupling of prostaglandin F2a receptor

We thank the reviewers for the positive assessments on the quality and importance of our works. Their constructive suggestions have helped us tremendously in revising our manuscript. In the following sections, we provide point-by-point responses to the comments by the two reviewers of our original paper. The reviewer's comments are in **black** and our responses are in **blue**. The changes in the manuscript are highlighted with **gray background**.

Point-by point response to Reviewer #1:

In this manuscript "Unique features of ligand-induced activation and G protein coupling of prostaglandin F2a receptor" by Canrong Wu, H. Eric Xu, and co-authors, they determined the structures of F2a receptor (FP) in complex with engineered Gq protein in the presence of three different agonists: endogenous ligand PGF2a, and synthetic ligand LTPA and TFPA. FP belongs to the prostaglandin receptor family of GPCRs playing pivotal roles in in regulating inflammation, allergy, intraocular pressure, and blood pressure. Through the cryoEM structures of FP, the authors reveal the ligand binding mode of the receptor, unique signature motif associated activation, and the engagement mode with the downstream G protein by combining structural analysis, mutagenesis, and IP1 cellular signaling assay. Notably, the authors provide structural framework governing the ligand-receptor selectivity. Carboxyl group of the alpha-chain and hydrophobic nature of the omega-chain of the ligand are crucial for the tight receptor binding, and F-ring for the receptor specificity. The findings presented in this study are of particular importance for the design of the ligand that selectively modulates the target receptor among the prostaglandin family with reduced aversive effect.

The overall writing in this report is well described.

I have several comments and questions as listed below.

Response: We thanks the reviewer for the positive comment on the importance of this work and the quality of the manuscript. The specific points are addressed below.

Having looked at the cryoEM density maps and the models, it seems that the density corresponding to the ligand is weak compared to the protein density surrounding the ligand for all three structures. At the low contour level, they all show up appropriately like Fig1a. This looks indicating low occupancy of the ligand. Given the high affinity of the ligands, it is hard to consider that the ligand falls off the pocket so easily. Do the authors have any comment?

Response: We thank the reviewer for the careful reading and point out this. As described in Supplementary Figure 1-3, we acquired both postprocessing maps and DeepEMhancer maps. In all three postprocessing maps, densities corresponding to three ligands are similar to the protein density (Figure R1). While in DeepEMhancer maps, the density corresponding to the ligand is weak compared to the protein density surrounding the ligand for all three structures. DeepEMhancer process were executed to reduce noise levels and obtain more detailed version of experimental maps for structural analysis (Sanchez-Garcia *et al.* Commun Biol , 2021). However, the densities of these ligands are possibly weakened after DeepEMhancer process (<https://github.com/rsanchezgarc/deepEMhancer/issues/11>), rather than caused by the ligand falls off the pocket. The atomic coordinates and the electron microscopy maps have been deposited in the Protein Data Bank (PDB) under accession number 8IUK, 8IUL and 8IUM and Electron Microscopy Data Bank (EMDB) accession number EMD-35724, EMD-35725 and EMD-35726 for the PGF2 α -FP-Gq and the LTPA-FP-Gq and the TFPA-FP-Gq complex, respectively.

Figure R1. Density map of three ligands in postprocesses maps (Contour levels = 0.016).

Of the three ligands, PGF2 α fails to form a hydrogen bond with S118. Close look at the omega-chain density of the ligand, I agree that the hydroxyl group of PGF2 α points away from S118, while that of LTPA and fluoride of TFPA points toward S118. The authors briefly mention around this as potentially triggered by the presence of the aromatic ring. Given that the ligand selectivity is one of the main topics of this manuscript and S118 is unique to FP, this geometry switch should be discussed in more detail possibly even in the main text.

Response: We thank the reviewer for this valuable suggestion. We have described more detail about the S118 mediated ligand selectivity in our revised manuscript.

Despite PGF2 α contains the hydroxyl group at the same position as LTPA that the presence of the aromatic ring and a lack of carbon to carbon double bonds in the ω -chain of LTPA may affect the geometry of the ligand, indicating that these modifications in ω -chain chain of PGAs may improve their selectivity to FP.

The authors claim that the FP possesses the unique activation mechanisms. Indeed, this receptor lacks both “PIF” and “E/DRY” motifs. However, the activation mechanism itself appears very similar to the classical mode: The agonist triggers a conformational change of the toggle switch “W6.48”, which is transmitted to the “PIF” or an equivalent motif, then the “E/DRY” or an equivalent motif opens up to allow opening of TM6. It appears too much overemphasizing to claim FP activation undergoes unique mechanism including the title of the manuscript. I would rather prefer emphasizing more on the ligand selectivity from the structural analyses of these ligands.

Response: We appreciate the reviewer for the careful reading and constructive suggestions. We have highlighted ligand selectivity in the title and abstract of revised manuscript as below.

Ligand-induced activation and G protein coupling of prostaglandin F_{2α} receptor

These structures revealed distinct features of FP within the lipid receptor family in terms of ligand binding selectivity, its receptor activation, and G protein coupling mechanisms, including activation in the absence of canonical PIF and ERY motifs and Gq coupling through direct interactions with receptor transmembrane helix 1 and intracellular loop 1.

The authors employed chimeric Gq protein for the structural study. It is described in the Methods section, but please clarify the boundary between Gs part and Gq part as well.

Response: We thank the reviewer for the careful reviewing. We have clarified the composition of the engineered Gα_q chimera in detail as below.

An engineered Gα_q chimera was generated based on the mini-Gas scaffold with its aa 1-18 replaced by corresponding sequences of Gα_{i1}, aa348-359 replaced by corresponding sequences of Gα_q, A33, H35, A87, V92, D107, V115, R137, N144, C151, F155, K158, V161, K163, D171 and D319 replaced by corresponding residues in Gα_q, designated as mGα_{s/q/iN}.

Line 247: “C134^{3.51} in the “ERC” motif moves inwards, forming a polar 247 interaction with T223^{5.59} in TM5” The geometry doesn’t look these residues form a polar interaction.

Response: We thank the reviewer for the careful reviewing. we agree the view of the reviewer and have changed this sentence to “C134^{3.51} in the “ERC” motif moves inwards, forming interactions with I222^{5.58} and T223^{5.59} in TM5.”

Supplemental Figure 7a: The colors of the label don’t match those of the curve.

Response: We thank the reviewer for the careful reviewing. We have corrected it in the revised manuscript.

Point-by point response to Reviewer #2:

Canrong Wu and colleagues have provided a revised version of the manuscript, “Unique features of ligand-induced activation and G protein coupling of prostaglandin F2a receptor”. The authors have appropriately addressed previous concerns and comments in a manner that is suitable for publication. The overall findings are well supported and described and will be of broad interest to the field.

Response: We thanks the reviewer for the positive comment on this work.